# A One Health Approach Metagenomic Study on Antimicrobial Resistance Traits of Canine Saliva

**DOI:** 10.3390/antibiotics14050433

**Published:** 2025-04-25

**Authors:** Adrienn Gréta Tóth, Darinka Lilla Tóth, Laura Remport, Imre Tóth, Tibor Németh, Attila Dubecz, Árpád V. Patai, Zsombor Wagenhoffer, László Makrai, Norbert Solymosi

**Affiliations:** 1Centre for Bioinformatics, University of Veterinary Medicine, 1078 Budapest, Hungary; toth.adrienn.greta@univet.hu; 2Institute for Animal Breeding, Nutrition and Laboratory Animal Science, University of Veterinary Medicine, 1078 Budapest, Hungary; wagenhoffer.zsombor@univet.hu; 3School of Biodiversity, One Health & Veterinary Medicine, University of Glasglow, Glasgow G12 8QQ, UK; darinka.lilla.toth@gmail.com; 4OnlyVet Veterinary Referral Center, 69800 Saint-Priest, France; remportlaura@gmail.com; 5Department of Operative Tecniques and Surgical Research, Faculty of Medicine, University of Debrecen, 4032 Debrecen, Hungary; drtothimre@bazmkorhaz.hu; 6Department of Thoracic Surgery, Borsod-Abaúj-Zemplén County Hospital and University Teaching Hospital, 3526 Miskolc, Hungary; 7Department and Clinic of Surgery and Ophthalmology, University of Veterinary Medicine, 1078 Budapest, Hungary; 8Department of Surgery, Paracelsus Medical University, 90419 Nuremberg, Germany; attila.dubecz@klinikum-nuernberg.de; 9Division of Interventional Gastroenterology, Department of Surgery, Transplantation and Gastroenterology, Semmelweis University, 1082 Budapest, Hungary; patai.arpad@semmelweis.hu; 10Autovakcina Ltd., 1171 Budapest, Hungary; autovakcina@gmail.com; 11Department of Physics of Complex Systems, Eötvös Loránd University, 1117 Budapest, Hungary

**Keywords:** antimicrobial resistance, bacteriome, resistome, mobilome, dog saliva

## Abstract

**Background:** According to the One Health concept, the physical proximity between pets and their owners facilitates the interspecies spread of bacteria including those that may harbor numerous antimicrobial resistance genes (ARGs). **Methods:** A shotgun sequencing metagenomic data-based bacteriome and resistome study of 1830 canine saliva samples was conducted considering the subsets of ARGs with higher public health risk, ESKAPE pathogen relatedness (*Enterococcus faecium*, *Staphylococcus aureus*, *Klebsiella pneumoniae*, *Acinetobacter baumannii*, *Pseudomonas aeruginosa* and *Enterobacter species*), and survey results on the physical and behavioral characteristics of the participating dogs. **Results:** A total of 318 ARG types achieved sufficiently high detection rates. These ARGs can affect 31 antibiotic drug classes through various resistance mechanisms. ARGs against tetracyclines, cephalosporins, and, interestingly, peptides appeared in the highest number of samples. Other Critically Important Antimicrobials (CIAs, WHO), such as aminoglycosides, fluoroquinolones, or macrolides, were among the drug classes most frequently affected by ARGs of higher public health risk and ESKAPE pathogen-related ARGs of higher public health risk. Several characteristics, including coat color, sterilization status, size, activity, or aggressiveness, were associated with statistically significant differences in ARG occurrence rates (*p* < 0.0500). **Conclusions:** Although the oral microbiome of pet owners is unknown, the One Health and public health implications of the close human–pet bonds and the factors potentially underlying the increase in salivary ARG numbers should be considered, particularly in light of the presence of ARGs affecting critically important drugs for human medicine.

## 1. Introduction

The interconnection of human, animal, and environmental habitats contributes to the evolution of one of the concerning global health issues, antimicrobial resistance (AMR). Media involved in AMR transmission pathways among physically connected habitats can be screened for their role in the growing appearance rates of AMR. According to the One Health concept, such a pathway is among humans and their companion animals.

After a decade-long rise in the number of companion animals in households, more than 50% of the U.S. population held a dog in 2020 [1,2,3]. Moreover, a change in the nature of human–pet bonds is also notable. According to the survey of the American Veterinary Medical Association, 70% of owners regard their pets as family members, 17% as companions, and only 3% as property [4]. This mindset enhances physical proximity (e.g., sleeping together, licking the face of the owner, etc.) among pets and their owners. Moreover, companion dogs typically receive veterinary healthcare services, which can include the administration of antibiotics. Additionally, dog bite cases are frequent, and tend to be committed by family pets rather than strays [5].

Physical contact among pets and owners serves as a platform for bacterium transfer, including ESKAPE pathogens (*Enterococcus faecium*, *Staphylococcus aureus*, *Klebsiella pneumoniae*, *Acinetobacter baumannii*, *Pseudomonas aeruginosa*, and *Enterobacter* spp.) listed by the World Health Organization (WHO). Along with bacteria, AMR determinants, including higher-public-health-risk antimicrobial resistance genes (ARGs) as categorized by Zhang et al. [6], can also be transferred. Their study provides information on the associations of qualitative and quantitative AMR characteristics (overall ARG counts, ARG counts associated with enhanced public health risk, and ESKAPE pathogen-related higher-public-health-risk ARG counts) and certain easily observable canine traits (e.g., sterilization status, breed, head shape, fur color, behavioral characteristics). These findings may highlight the consequences of our companion animal-keeping behavior and help better understand the One Health impact of the dog–owner coexistence on the appearance and spread of AMR.

Our aim was to obtain information on the associations of three antimicrobial resistance gene (ARG) sets (all detected ARGs, higher-public-health-risk ARGs [6], and ESKAPE pathogen-related higher-public-health-risk ARGs) in canine saliva alongside certain easily observable canine traits (e.g., sterilization status, breed, head shape, fur color, behavioral properties) that may help better understand the One Health impact of the dog–owner coexistence and other companion animal-keeping behavior on the appearance and spread of AMR. For this purpose, canine shotgun metagenonomic data from the study of Morrill et al. [7] were used. The genomic dataset used was linked to responses from questionnaires consisting of 118 questions on canine characteristics and mostly behavioral measures, which were answered by the owners of the sampled dogs. ARG abundances were analyzed based on phenotypical groups formed based on these answers.

## 2. Results

Results obtained from the bioinformatic analysis of the 1830 canine saliva samples’ metagenome sequencing data are articulated as follows. After the presentation of the most frequently detected bacterial taxa (bacteriome) associated with ARGs and the set of identified ARGs (resistome), physical and behavioral traits potentially related to the AMR determinants in the canine salivary bacteriome are detailed. Results with potential biomedical significance are presented in the following sections. Further results are presented in the Appendix A.

### 2.1. Bacteriome

By the taxonomic classification, 126 genera were identified and associated with ARGs in 1682 of 1830 samples (91.9%). ARG-associated bacterial genera that appeared in at least 2% of the samples were *Porphyromonas* (33.1%), *Pasteurella* (31.4%), *Frederiksenia* (23.2%), *Pseudomonas* (23.0), *Conchiformibius* (21.7%), *Bacteroides* (10%), *Glaesserella* (9.5%), *Wielerella* (8.7%), *Riemerella* (8.7%), *Escherichia* (8.6%), *Prevotella* (7.3%), *Capnocytophaga* (6.8%), *Klebsiella* (5.2%), *Alistipes* (4.7%), *Lelliottia* (4.5%), *Mannheimia* (4.0%), *Histophilus* (3.9%), *Streptococcus* (3.8%), *Serratia* (3.7%), *Haemophilus* (3.1%), *Enterobacter* (2.6%), *Clostridioides* (2.4%), and *Parabacteroides* (2.4%). The numbers of samples in which these bacterial genera appeared are demonstrated in Figure 1.

### 2.2. Resistome

In setting 90% length and 90% base identity thresholds for ARG identification, 1682 samples were associated with ARGs in general, although only 1679 of these samples contained genes that can have antimicrobial effects. Three genes, namely *CRP, nalC*, and *nalD* that were identified in the samples based on the CARD database were ARG regulators or repressors. Thus, *CRP, nalC*, and *nalD* were excluded from the analysis. The highest number of ARG types (65) was found in BioSample SAMN16779058 and SAMN16778891 (57), followed by SAMN16778736 and SAMN16779288 (55). The mean of ARG hits per sample reached 4.3, and the median was 3. In total, 318 ARG types met the criteria of having 90% coverage and 90% sequential identity. The most common ARGs were *Escherichia coli EF-Tu* mutants conferring resistance to Pulvomycin, *pgpB*, and *CfxA2* genes, detected in 1002, 953, and 609 samples, respectively. ARG hits of higher public health significance were detected in 1033 samples. The most public health risk hits were discovered in Biosample SAMN16779288 (35), SAMN16779058 (30), and SAMN16778891 in parallel with SAMN16777653 (23). The average number of higher-public-health-risk ARG hits per sample was 2.3, while the median was 1. Of the 65 detected ARGs of higher public health risk, the most common were *CfxA2*, *ErmF*, and *APH(6)-Id*, appearing in 609, 202, and 153 samples, respectively. ESKAPE pathogen-related ARG hits of higher public health risk were identified in 63 samples. Biosample SAMN16778736 contained the most hits of this type (15), followed by SAMN16777571 (14) and SAMN16778717 (13). The mean of ESKAPE pathogen-related higher-public-health-risk ARG hits per sample amounted to 3.3, and the median to 2. All in all, 13 types of ARG hits of public health risk were detected in ESKAPE pathogens. Of these, the most common were *OXA-2* (18 samples), *APH(6)-Id* (17 samples), and *sul1* (15 samples). In total, 20.4% of all ARG types were higher-public-health-risk ARGs. In examining ESKAPE pathogen-related higher-public-health-risk ARG types, proportions amounted to 4% considering all ARG hits and 20% of higher-public-health-risk ARG types. ARGs appearing in more than 10 samples together with higher-public-health-risk ARGs and higher-public-health-risk ARGs associated with ESKAPE pathogens that appeared in at least five samples are shown in Figure 2. The full list of ARGs and the numbers of samples in which they appeared can be seen in Appendix A.

Drug classes that could potentially be affected by the presence of the detected ARGs are presented in Figure 3, considering all ARGs, higher-public-health-risk ARGs, and higher-public-health-risk ARGs in ESKAPE pathogens. ARGs detected in the samples can potentially affect 31 antibiotic classes. The most commonly affected antimicrobial groups were tetracyclines (1650 samples), cephalosporins (1125 samples), peptides (1105 samples), and penams (1095 samples) considering all detected ARGs. In the case of higher-public-health-risk ARGs, cephamycin resistance genes were the most frequent (754 samples), followed by macrolides (529 samples) and amynoglycosides (508 samples). In higher-public-health-risk ARGs in ESKAPE pathogens, amynoglycosides (61 samples), fluoroquinolones (59 samples), phenicols (54 samples), penams (53 samples), and tetracyclines (53 samples) appeared the most often. Considering all three groups (all ARGs, higher-public-health-risk ARGs, higher-public-health-risk ARGs in ESKAPE pathogens), pleuromutilin and oxazolidinone resistance appeared in the fewest number of samples. The exact sample numbers by antibiotic groups can be seen in Appendix A.

The proportion of resistance mechanisms was calculated based on the diversity of all ARGs, higher-public-health-risk ARGs, and higher-public-health-risk ARGs in ESKAPE pathogens. The results are shown in Figure 4. The dominant mechanism of identified ARGs was antibiotic inactivation (2669 occasions), antibiotic target alteration (2403 occasions), and antibiotic efflux (973 occasions). In the case of higher-public-health-risk ARGs, the top AMR mechanisms were antibiotic inactivation (1304 occasions), antibiotic efflux (533 occasions), and antibiotic target alteration (292 occasions). Higher-public-health-risk ARGs identified in ESKAPE pathogens were in most cases associated with antibiotic efflux (110 occasions), antibiotic inactivation, and antibiotic target replacement (25 occasions).

### 2.3. Basic Characteristics, Physical Traits, and Behavioral Characteristics Associated with ARGs

The proportions of statistically significant and nonsignificant results of ARG abundances and traits or answers of the surveys differed in the examined animal groups. The proportions of questions with significant (p≤0.05) and nonsignificant statistical test results by question types (see Appendix A) and by approaches A, B or C are presented in Figure 5.

#### 2.3.1. Basic Characteristics and Physical Traits

In analyzing the associations of the numbers of ARG-positive and ARG-negative samples in the light of basic characteristics, the results presented in Table 1 were obtained. Considering all approaches, three out of six questions had statistically significant results.

The associations of the ARG content of the saliva samples and physical traits such as the fur color, fur type, tail type, ear shape, eye characteristics, fur length, and fur texture are presented in Appendix A. By certain questions, multiple comparisons of the trait options were performed.

#### 2.3.2. Behavioral Traits

Due to the high number of behavioral traits included in the original questionnaire, only the most significant results are mentioned in Section 2. See Appendix A for full behavioral result tables.

For questions that could have been answered with Never or Not never (Always, Often, Sometimes, Rarely), two statistically significant results were found. The statistical test results for all questions of this answer category are presented in Appendix A.

In the comparison of the number of ARG-positive and ARG-negative samples for questions with answer groups Agree (Strongly agree, Agree)/Disagree (Strongly disagree, disagree), several statistically significant results were obtained. Statistical test results for all questions, including those with an unclear biomedical significance, are presented in Appendix A.

Comparing the answer groups of Agree (Strongly agree, Agree); Neither agree, nor disagree; and Disagree (Strongly disagree, disagree) gave similar but not exactly identical results, which can be found in Appendix A.

## 3. Discussion

Throughout this study, several AMR-related aspects of the canine oral cavity and saliva were revealed.

Apart from the genus of Wielerella, all ARG-associated bacterial genera were identified in the canine oral cavity, or along the gastrointestinal tract previously [8,9,10].

After the bioinformatic analysis conducted based on reference data from the Comprehensive Antimicrobial Resistance Database (CARD), a high number of AMR determinants were identified in the canine saliva samples. AMR determinants stored in the CARD are all associated with AMR [11]. Several genes are directly responsible for the appearance of AMR (e.g., *dfra14*, *dfra20*, *sul1*, *sul2*, *SRT-2*, *SRT-3*, *SAT-4*, *tet* family, *ROB* family, *APH* family, OXA family, etc.) [11]. However, several ARGs mentioned in Section 2 are essential for cell viability and eventually also cause AMR. Certain ARGs (e.g., *mel*, *msbA*) are sequence homologs of previously known ARGs that can be responsible for AMR themselves [11], and some encode for intrinsic, non-transferable AMR characteristics (e.g., *mexA*, *mexB*, *mexC*, *mexE* [12]. In other cases, the mutation of an intrinsic, originally non-AMR encoding gene causes specific antibiotic effects (e.g., *Escherichia coli EF-Tu* mutants conferring resistance to Pulvomycin, *Escherichia coli UhpT* with mutation conferring resistance to fosfomycin). Other genes that indirectly induce or regulate AMR facilitators (e.g., *ArmR*, *baeR*, *H-NS*, *marA*, *ramA*) [11] are also inevitable in the phenotypical appearance of AMR. Furthermore, genes encoding the members of complexes that realize AMR or broader-spectrum AMR were also detected (e.g., *mdtB, mdtC, mex* family) [11]. In the case of the abovementioned mdtBC multidrug transporter, even the regulator, *baeR*, was identified in many samples [11]. In other cases, only subunits or functional cooperators of AMR determinants complexes (e.g., efflux pumps) were identified (e.g., *qacL*, *oprM*, *oprN*, *OpmB*, *opmE*) [11]. Although associated with AMR, the presence of certain determinants, such as *CRP*, *nalC*, and *nalD*, is also beneficial due to the repression or modulation of multidrug resistance. Despite the presence of *CRP*, *nalC*, and *nalD* both in the CARD database and in numerous samples, these genes were left out of further analysis steps due to their anti-AMR effect [11]. To enhance public health risk, Zhang et al. [6] used four indicators (higher human accessibility, mobility, pathogenicity, and clinical availability) to identify higher-public-health-risk ARGs. Based on the indicators, higher-public-health-risk ARGs and higher-public-health-risk ARGs detected in ESKAPE pathogens have a strong AMR potential and an enhanced public health significance.

Putting the affected antimicrobials into perspective, according to the World Health Organization (WHO)’s review, Critically Important Antimicrobials (CIAs) (divided into Highest-Priority Critically Important Antimicrobials (Category 1) and High-Priority Critically Important Antimicrobials (Category 2)) are in need of the most urgent AMR risk management steps [13]. Among the overall ARG set, the most hits may affect tetracyclines, cephalosporins, peptides, and penams. Antibiotic compounds from these groups, such as amoxicillin–clavulanate, first-generation cephalosporins, or doxycycline are the flagships of routine antibiotic administration in small animal veterinary practices [14]. However, some Highest-Priority CIAs, namely third-, fourth-, and fifth-generation cephalosporins, glycopeptides, and polymyxins are also members of these groups. Members of the *van* gene family, which confer glycopeptide resistance- [15] and polimyxin resistance-associated genes, including *eptB,* plasmid-encoded *MCR-9.1* [16], *mtrA*, *PmrF*, *AcrF*, *AcrE*, *KpnH*, *KpnF*, *KpnG*, and *KpnE* [17,18], were also identified in the canine saliva samples. However, third- and fourth-generation cephalosporin resistance-associated *CTX-M* gene family members were not found in the samples, while determinants from other extended-spectrum beta-lactamase gene families, such as *SHV (SHV-1, SHV-11)* and *TEM (TEM-135, TEM-178)*, were detected [19]. A regulatory member of the Methicillin-Resistant *Staphylococcus aureus* and *Staphylococcus epidermidis*-associated *mec* family (*mecI*) was also identified [20]. Regarding the remaining Highest-Priority CIAs (Macrolides, ketolides, Quinolones), the number of resistance determinants against them was so high that they were presented to become the second most and the most frequently affected drug classes considering higher-public-health-risk ARGs and higher-public-health-risk ARGs in ESKAPE pathogens, respectively. High-Priority CIA aminoglycosides were the third most affected group by higher-public-health-risk ARGs. Furthermore, 59 AMR determinant types appeared against carbapenems and other penems, which are also Category 2 CIAs.

The shift in relative abundances towards antibiotic efflux, a mechanism type capable of affecting multiple drug classes parallelly in ESKAPE pathogen-related higher-health-risk ARGs is because multidrug resistance exhibited by ESKAPE pathogens is widespread [21].

Relatively more physical trait-related questions had significant results that may be associated with the fact that physical traits are objectively observable, while behavioral traits can be determined rather subjectively. Doliocephalic dogs are slightly more often ARG-positive than brachycephalics. In contrast, strong evidence of brachycephalic breeds being less healthy has been provided [22], which could possibly lead to more visits to the veterinarian. However, popular brachycephalic breeds, such as the French bulldog have a shorter life expectancy [23] that can contribute to the balance. Considering certain inherited disorders, recently bred, similar canine lineages appear to be more susceptible [24]. The higher number of ESKAPE pathogen-related higher-health-risk ARG-positive samples may either be due to this or owner habits regarding the frequencies of veterinary visits with more expensive dogs. Nonetheless, Singleton et al. found that antibiotic use was less likely in sterilized dogs [25], as higher-health-risk ARGs were significantly less frequently detected in intact canines. The significantly higher number of higher-public-health-risk ARGs in under knee-high dogs might be associated with these breeds sharing the same environment. Regarding the colors, there seems to be a clear trend of more ARG-positive samples among white dogs compared to other colors, even though the question regarding the white fur ratio of the dogs suggests the contrary. The most possible reason for this inconsistency can be seen in Appendix A, where breed color and breed whiteness associations are presented. The majority of positive answers to the question regarding the white fur ratio of the dogs, and namely if the dog was all white, came from owners of golden retrievers and Labrador retrievers (Appendix A). Even though these dogs often have a light coat, a yellow coat is far more frequent than pure white. By the questions regarding the exact color of the same breeds (Appendix A), few answered that their Labradors or golden retrievers are white. Based on this, we can assume that the question regarding the coat whiteness ratio of dogs was considered to apply to the ratio of light fur to dark fur. However, genes responsible for white fur also cause severe phenotypes, including extensive depigmentation, hearing loss, and acute eye and bone disorders [26] that may facilitate more frequent veterinary visits. The finding that a shift to ESKAPE pathogen-related high-risk ARG-positive samples also appeared in dogs with heterochromia is in line with the findings of white fur color. Breeds with the highest number of answers of white fur color (American pit bull terriers, border collies, huskies, etc.) are also prone to having different eye colors [27,28]. Moreover, when white, merle, or gray dogs were examined together, higher ARG positivity was detected. In merle and gray dogs, heterochromia is also common [28], and certain hereditary diseases are also more prevalent (ocular and auditory anomalies) [29,30]. Even though long pendant ears may facilitate external ear canal infections, no related significant effect could was observed in the number of ARG-positive samples. In contrast, a rise in the number of ESKAPE pathogen-related higher-health-risk ARG-positives among canines with long fur was noticeable.

The evaluation of behavioral findings is very difficult. Nevertheless, a pattern that seems to appear considering all significant results is that dogs that are more active less often harbored ARGs, higher-health-risk ARGs, or ESKAPE pathogen-related higher-health-risk ARGs in their saliva samples. Another possible trend is that dogs that are less aggressive in general are more often ARG- or higher-health-risk ARG-positive. The explanation of this phenomenon is unclear but might derive from the fact that less aggressive dogs are more favorable as pets and are brought to veterinarians more regularly due to the closer human bonds.

Interestingly, the proportion of statistically significant results was the highest based on the “motor pattern” of all question types with approach B, while no significant results of this question type appeared with any other approaches (A or C). Questions with statistically significant results of this type aimed to reveal the playfulness of dogs. According to this, decreased playfulness might be especially characteristic for elevated higher-public-health-risk ARG presence in canine salivary bacteria. By approach A and C, the “physical trait” question type had the highest proportion of significant test results. This finding may be associated with the fact that physical traits are easier to observe objectively than behavioral traits.

## 4. Materials and Methods

### 4.1. Data

Canine salivary genomic datasets were obtained from the National Center for Biotechnology Information (NCBI) Sequence Read Archive (SRA) repository. The BioProject used is stored under accession number PRJNA675863 submitted by the Broad Institute, Darwin’s Ark project by Morrill et al. [7] (https://darwinsark.org/, accessed on 14 April 2025). The authors had no access to information that could identify individual participants during or after data collection. The original study of Morrill et al. includes the animal study protocols for saliva and blood collection from dogs and the consent from the owners for the participation of their animals in the original study [7]. The project contained the shotgun metagenomic sequencing data of canine saliva samples to investigate polymorphisms associated with complex canine traits [7]. The BioProject contained short read data from 2215 canine saliva samples, out of which 1830 were used in this study. The selection process was based on the completeness of the metadata information of the samples. Only samples with full metadata information were retained. The metadata used were derived from the Broad Institute’s Darwin’s Ark project survey dataset (https://datadryad.org/stash/dataset/doi:10.5061/dryad.g4f4qrfr0, accessed on 8 December 2023). Throughout the analysis steps, the presence of any ARGs, higher-public-health-risk ARGs, and ESKAPE pathogen-related higher-public-health-risk ARGs within the metagenomic samples was signed with “A”, “B”, and “C”, respectively. The survey that gave the basis for the trait associated AMR studies was filled voluntarily by dog owners. Out of 12 questionnaires, 11 surveyed canine behavior, and 1 focused on physical traits. Breed or suspected breeds, age, sex, and sterilization status were collected initially [7]. Samples were analyzed based on their unique darwinsark_ids. However, in case of sex, only those BioSample identifiers were included in the analyses that could have been associated with unique darwinsark_id identifiers or with no darwinsark_id identifiers. All questions in the survey of Morrill et al. [7] were included in the analysis workflow. However, only questions with possible biomedical explanations are presented in Section 2. All other question results can be found in the Appendix A.

### 4.2. Bioinformatic and Statistical Analysis

After quality control, the trimming and filtering of the raw short reads were performed with TrimGalore (v.0.6.6, https://github.com/FelixKrueger/TrimGalore, accessed on 8 December 2023), with a quality threshold of 20. More than 50 bp long reads were taxonomically classified using Kraken2 (v2.1.1) [31] using a database created (4 June 2022) based on NCBI RefSeq complete archaeal, bacterial, viral, and plant genomes. For precise species assignment, a confidence parameter of 0.5 was used. The taxon classification results were processed in the R-environment [32] using the packages phyloseq (v1.36.0) [33] and microbiome (v1.14.0) [34]. Bacterial reads were assembled to contigs with MEGAHIT (v1.2.9) [35] using default settings. Kraken2 was used again for the taxonomic classification of the contigs based on the same database mentioned above. Possible open reading frames (ORFs) were extracted from the contigs with Prodigal (v2.6.3) [36]. Amino acid sequences translated from the ORFs were aligned to the reference ARG sequences of the Comprehensive Antibiotic Resistance Database (CARD, v.3.1.3) [11,37] with Resistance Gene Identifier (RGI, v5.2.0) with Diamond [38]. ORFs classified as “Perfect” or “Strict” were further filtered with 90% identity and 90% coverage to the reference sequences. The public health risk categorization of the ARGs was based on a publication of Zhang et al. [6]; they used four indicators to categorize ARGs with a higher public health risk: higher human accessibility, mobility, pathogenicity, and clinical availability. Human accessibility was based on the transfer likelihood of the ARG from the environment to the human microbiota. Mobility was defined by the relatedness of mobile genetic elements to the ARG. Human pathogenicity was based on appearance rates in human pathogens compared to that in non-pathogenic bacteria. Clinical availability was evaluated according to the use and clinical relevance of the affected antimicrobial agents. The independence of ARG frequencies on various grouping variables was tested using Fisher’s exact test [39]. A *p*-value of <0.05 was considered statistically significant. All further data management procedures, analyses, and plotting were performed in the R-environment (v4.2.1) [32].

## 5. Conclusions

To fully understand the One Health implications of the results obtained, further studies assessing the relatedness of the skin microbiome of pets and their owners would be required. Such investigations would explain pathways of microbial genetic material exchange and antimicrobial resistance spreading and transmission between humans and animals. However, disturbing findings, such as the high detection rate of peptides and other CIAs, indeed draw attention to the potential One Health and public health significance of human–pet proximity.

## Figures and Tables

**Figure 1 antibiotics-14-00433-f001:**
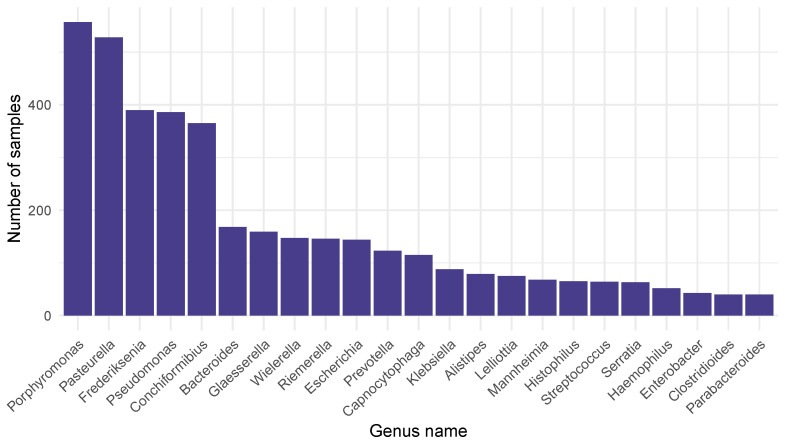
Bacterial genera associated with AMR determinants that appeared in at least 2% of all 1682 samples, with the number of samples in which they were detected.

**Figure 2 antibiotics-14-00433-f002:**
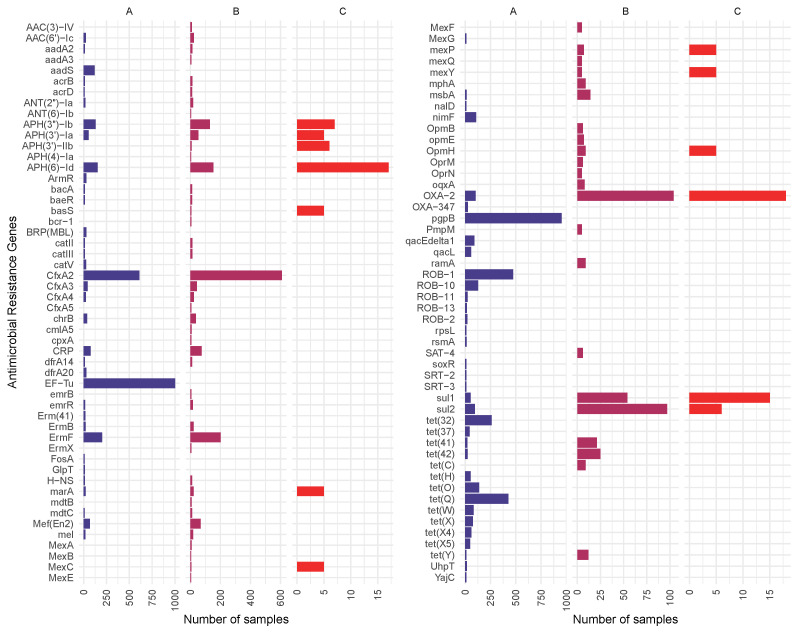
ARGs detected in more than 10 samples (**A**), higher-public-health-risk ARGs detected in at least 5 samples (**B**), and higher-public-health-risk ARGs derived from ESKAPE pathogens detected in at least 5 samples (**C**). *EF-Tu* is the abbreviation for *Escherichia coli EF-Tu* mutants conferring resistance to Pulvomycin, *UhpT* for *E. coli UhpT* with mutations conferring resistance to fosfomycin, *GlpT* for *E. coli GlpT* with mutations conferring resistance to fosfomycin, *soxR* for *Pseudomonas aeruginosa soxR*, and *rpsL* for *Mycobacterium tuberculosis rpsL* mutations conferring resistance to streptomycin.

**Figure 3 antibiotics-14-00433-f003:**
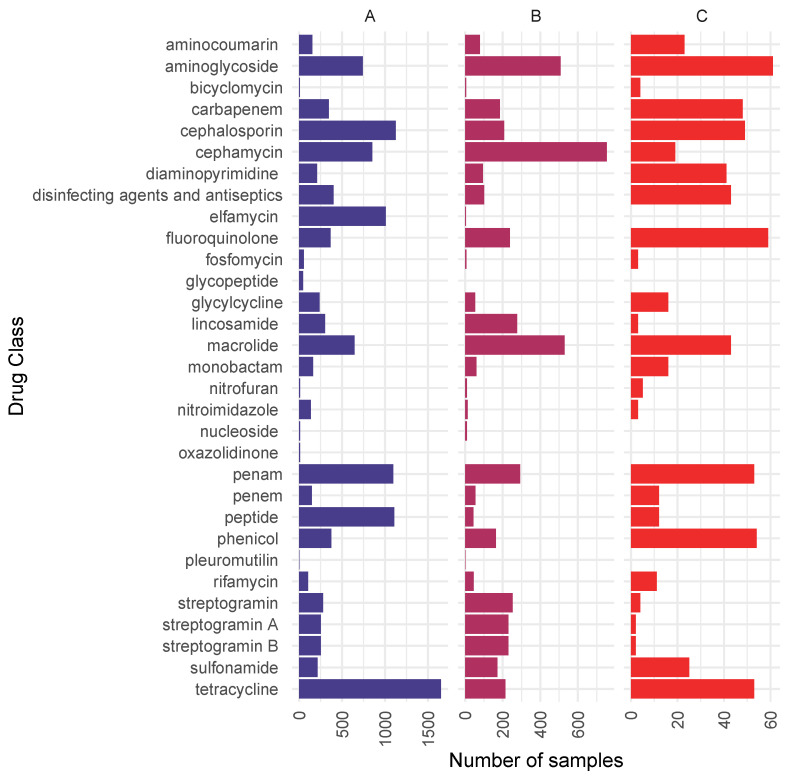
Antibiotic groups against which ARGs were detected in any metagenomic samples (**A**), antibiotic groups against which higher-public-health-risk ARGs were detected in any metagenomic samples (**B**), and antibiotic groups against which higher-public-health-risk ARGs appeared in ESKAPE pathogens (**C**). The number of samples in which ARGs against the presented antibiotic groups were detected is presented on the horizontal axis. Antibiotic compounds affected by multidrug resistance are displayed separately.

**Figure 4 antibiotics-14-00433-f004:**
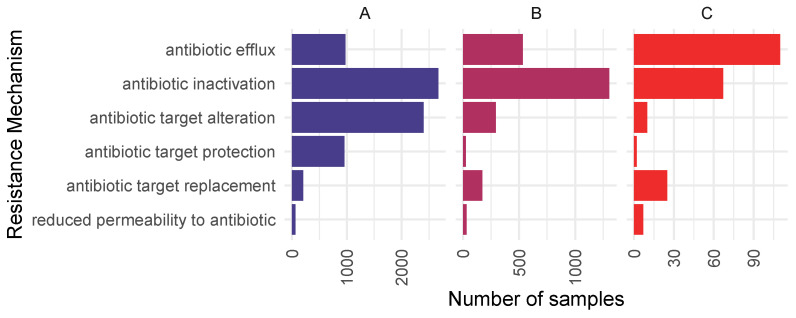
Antimicrobial resistance mechanism abundances of ARGs detected in all metagenomic samples (**A**), antimicrobial resistance mechanism abundances of higher-public-health-risk ARGs detected in any metagenomic samples (**B**), and antimicrobial resistance mechanism abundances of higher-public-health-risk ARGs in ESKAPE pathogens detected in the metagenomic samples (**C**). The number of occasions when ARGs with the given antimicrobial resistance mechanisms were detected is presented on the horizontal axis.

**Figure 5 antibiotics-14-00433-f005:**
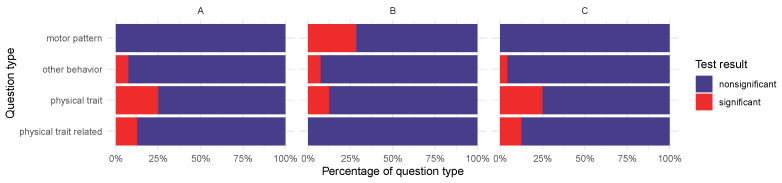
Proportions of significant (p≤0.05) and nonsignificant associations by question groups (see Appendix A) and approaches (ARGs detected in all metagenomic samples (**A**), higher-public-health-risk ARGs detected in any metagenomic samples (**B**), and ESKAPE pathogen-related higher-public-health-risk ARGs in the metagenomic samples (**C**)).

**Table 1 antibiotics-14-00433-t001:** The number of samples containing ARGs (ARG+) and lacking ARGs (ARG−) associated with the following basic characteristics and traits of dogs: head shape, origin, sex, sterilization status, breed, environment. “Group 1” and “Group 2” represent the order of traits indicated in the “Characteristic” column of the table. In column “Approach”, “A” indicates ARGs detected in any canine metagenomic samples, “B” is for higher-public-health-risk ARGs detected in any canine metagenomic samples, and “C” stands for higher-public-health-risk ARGs detected in ESKAPE pathogens. No stars is: *p*-value > 0.05 (not statistically significant), one star (*): *p*-value ≤ 0.05 (statistically significant at the 5% level), two stars (**): *p*-value ≤ 0.01 (statistically significant at the 1% level).

Characteristic	Approach	Number of Samples				OR (95%CI)	*p*-Value
Group 1/Group 2		Group 1		Group 2			
		ARG+	ARG−	ARG+	ARG−		
Brachycephal/dolio-, mesocephal	A	71	12	606	50	0.49 (0.24–1.06)	0.055
	B	44	39	370	286	0.87 (0.54–1.42)	0.560
	C	7	105	29	598	1.37 (0.50–3.31)	0.473
Breeder/rescue, shelter	A	286	28	1013	86	0.87 (0.55–1.41)	0.557
	B	172	142	628	472	0.91 (0.70–1.18)	0.478
	C	13	301	33	1067	1.40 (0.67–2.77)	0.366
Females/males	A	827	68	852	83	1.18 (0.84–1.68)	0.350
	B	511	384	519	416	1.13 (0.76–1.70)	0.561
	C	27	828	33	861	0.85 (0.49–1.47)	0.600
Intact/sterilised	A	87	10	1515	131	0.75 (0.38–1.66)	0.440
	B	44	53	938	709	0.63 (0.41–0.97)	0.027 *
	C	2	95	58	1589	0.58 (0.07–2.24)	0.771
Purebred/mutt	A	677	62	935	80	0.93 (0.65–1.34)	0.723
	B	414	325	577	439	0.97 (0.80–1.18)	0.770
	C	36	703	25	991	2.03 (1.17–3.56)	0.008 **
Rural, suburban/urban	A	575	46	837	76	1.13 (0.76–1.70)	0.564
	B	346	275	515	399	0.97 (0.79–1.20)	0.834
	C	23	598	31	883	1.10 (0.60–1.96)	0.779

## Data Availability

The short-read data of samples are publicly available and accessible through PRJNA675863 from the NCBI Sequence Read Archive (SRA).

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
