# Peer review of "A One Health Approach Metagenomic Study on Antimicrobial Resistance Traits of Canine Saliva"

_antibiotics, 2025, doi:10.3390/antibiotics14050433_

Round 1

Reviewer 1 Report

Comments and Suggestions for Authors

Dear Author,

I would like to thank you for giving me the chance to review this manuscript titled ( A One Health approach metagenomic study on the antimicrobial resistance traits of canine saliva).

I have go through this manuscript and carefully evaluated it, and found its well written, and structured. Studying antibody resistance in humans and animals is of great importance due to the increased spread of antimicrobial resistance (AMR) among zoonotic pathogens. Furthermore, ESKAPE pathogens have gained high attention recently due to their high antibiotic resistance. Thus, understanding the spread of these pathogens, along with AMR and antibiotic resistance genes (ARGs), is of great importance for fighting this global health issue.

This study added a huge amount of Data associated with the diversity and prevalence of antibiotic resistance genes (ARGs) and ESKAPE pathogens in dogs (the most common pets worldwide) saliva. This study is supported by a comprehensive analysis of huge genetic data ARG distribution in 1830 dogs, revealing the fact that microorganisms present in canine saliva are reservoirs of ARGs that could spread to human pathogens. The authors used the needed software to perform the analysis to fulfill the aim of this study.

The manuscript could be published after minor revisions to enhance its clarity.enhance its clarity.

Introduction:

  • Highlights the direct relationship between AMR transmission from dogs to owners and there role in increasing infection risk.

The interconnection of human, animal and environmental habitats contributes to22
the evolution of one a concerning global health issues, antimicrobial resistance (AMR) (Could be modified to one of the concerning)

The improvement that can be added to this kind of study had been mentioned in the conclusion (To fully understand the One Health implications of the results arisen, further studies assessing the relatedness of the skin microbiome of pets and their owners would be required) such investigation would explain pathways of microbial genetic material exchange and antimicrobial resistance (AMR) spreading and transmission between humans and animals.

The conclusion is concise, but it is professionally and effectively written, . A few of the references are very old, yet they are appropriate with all other references.

Regards,

Author Response

Comment: Highlights the direct relationship between AMR transmission from dogs to owners and there role in increasing infection risk.

Response: Thank you for the suggestion, as far as the link is plausible based on our data, we have stressed this in the manuscript.

Comment: Could be modified to one of the concerning

Response: We have edited te text.

Comment: The improvement that can be added to this kind of study had been mentioned in the conclusion (To fully understand the One Health implications of the results arisen, further studies assessing the relatedness of the skin microbiome of pets and their owners would be required) such investigation would explain pathways of microbial genetic material exchange and antimicrobial resistance (AMR) spreading and transmission between humans and animals.

Response: We have added the proposed text. 

Reviewer 2 Report

Comments and Suggestions for Authors

Please refer to the attached for minor comments and suggestions.

Author Response

Comment: More clarity on why each trait or characteristic was of interest or could be relevant to the study factors in question

Response: Thank You for the comment. As indicated in the manuscript, the original survey-based dataset was used to articulate the ARG-traits/characteristic associations. However, only questions with possible biomedical explanations are presented in the Results section. The rest of the results are also included in the manuscript, however, among the supplementary materials.  

Comment: Line 23: delete ‘one’ = of one a concerning...
Response: The sentence was modified according the comments of Referee 1.

Comment: Line 28: the word ‘nature’ may be more accurate than ‘quality’ here
Response: We have changed the word.

Comment: Line 30: add (AVMA), reword: ‘...70% of pet owners consider their pets as family members...’ to ‘70% of owners regard their pets as family members’
Response: We have reworded the text. 

Comment: Lines 33 to 35, suggest rewording to: Moreover, companion dogs typically receive veterinary healthcare services, which can include administration of antibiotics. When dog bite cases arise, they tend to be committed by family pets rather than strays.

Response: We have reworded the text. 

Comment: Line 40: ARGs as characterized (add ‘as’)
Response: We have added the word. 

Comment: Split sentence that spans lines 40 – 48 into 2.
Response: We have splitted the sentence.

Comment: Line 51: replace ‘and’ with ‘alongside’ = saliva alongside certain...
Response: We have changed the word.

Comment: Line 55, suggest rewording slightly to: ‘...that may help better understand the OneHealth impact of the dog-owner coexistence and other companion animal keeping behaviours on the appearance and spread of AMR
Response: We have reworded the text. 

Comment: Line 58: what does the 118 apply to specifically?
Response: We made clarification in the text. The figure is for the questions with answers in the dataset.

Comment: Line 76: change ‘kept’ to ‘retained’
Response: We have changed the word.

Comment: Line 82: questionnaires or questions?
Response: The number of questionnaires.

Comment: Line 89-90, reword to: All other question results can be found in the Supplementary materials.
Response: We have reworded the text. 

Comment: Lines 128-129: the use of ‘could have been’ seems a bit uncertain. Do the authors in fact mean ‘could be’?
Response: We have clarified it, using only 'have been' without could. 

Comment: Line 129: I suggest re-referencing the 1830 in relation to the 1682 (either 1682 of the 1830 samples or 1682 (91.9%).
Response: We have reworded the text. 

Comment: Similarly, in Figure 1, how do the 1682 samples correspond to the total 1830 samples?
Response: Thank you for the question. While a total number of 1830 metagenomic samples were involved in the study initially, only 1682 of these could have been associated with ARGs with the ARG identification thresholds applied (90% length and 90% base identitity). This is indicated in the Results. 

Comment: Line 136: ‘n’ missing from the word ‘demostrated’
Response: We have edited the text. 

Comment: Line 139: again, how does 1679 correspond to the total number of samples
Response: Earlier it is described not all sample contained ARG, and in the following sentence we show some ARGs are not 'against' antibiotics as an argumentation of the difference. 

Comment: Line 167: seen at in table...
Response: We have edited the text. 

Comment: Figure 2 caption: what is the significance of detection in more than 10 samples, and in 5 samples?
Response: Due to the high number of detected ARG types, we decided to set the visualization threshold to at least 10 samples by all ARGs. At the same time, due to the lower ARG type count types by higher public health risk ARGs and higher public health risk ARGs deriving from ESKAPE pathogens more results could have been visualized. Thus the limits were set to keep a rational balance in the visualization. However, the whole set of results is presented in Supplementary table 4.

Comment: Lines 180-181: replace ‘in the lowest number of samples’ with ‘in the fewest samples’
Response: We have repleced.

Comment: Line 185: IN not ON figure...
Response: We have changed the word.

Comment: Line 199: dash or not after positive and negative?
Response: Thank You for noticing this, we have corrected this issue.

Comment: Figure 5 caption: what does ‘question groups’ refer to?
Response: Thank You for pointing this out. The question groups are presented in Supplementary table 9. This piece of information is now mentioned in the text and the caption as well.

Comment: Second line, Table 1 caption: traits and characteristics
Response: We have edited the text.

Comment: Line 202: remove the comma between traits and such ‘...traits such...’
Response: We have edited the text.

Comment: Line 203: size of the dog?
Response: Thank You for the comment, the necessary changes were made in the text.

Comment: Line 293: add be – may either be
Response: We have added it. 

Comment: Lines 294 – 295: the word ‘that’ is in duplicate
Response: We have deleted the dubplication.

Comment: Lines 296-298: I would urge caution and rewording regarding this statement ‘The reason might be that the owners of sterilized animals are more likely to seek veterinary care due to closer bonds or more responsibility towards the animals’ because numerous owners do not sterilize their animals precisely for health concerns
Response: Thank You for the comment. The related sentence was removed from the manuscript.

Comment: Similarly, Lines 298-300: ‘The significantly higher number of higher public health risk ARGs under knee-high can be underlain by the phenomenon that toy/pet dogs have a closer bond to their owners and share the same environment.’ Is overly generalized.
Response: Thank You for the comment. The above referred sentence is removed from the manuscript.